# Exploring the Critical Driving Forces and Strategy Adoption Paths of Professional Competency Development for Various Emergency Physicians Based on the Hybrid MCDM Approach

**DOI:** 10.3390/healthcare11040471

**Published:** 2023-02-06

**Authors:** Meng-Wei Chang, Chia-Te Kung, Shan-Fu Yu, Hui-Ting Wang, Chia-Li Lin

**Affiliations:** 1Department of Emergency Medicine, Kaohsiung Chang Gung Memorial Hospital, Kaohsiung 833, Taiwan; 2Chang Gung Medical Education Research Centre (CG-MERC), Taoyuan 333, Taiwan; 3Graduate Institute of Adult Education, National Kaohsiung Normal University, Kaohsiung 802, Taiwan; 4School of Medicine, College of Medicine, Chang Gung University, Taoyuan 333, Taiwan; 5Division of Rheumatology, Allergy, and Immunology, Department of Internal Medicine, Kaohsiung Chang Gung Memorial Hospital, Kaohsiung 833, Taiwan; 6Division of Rheumatology, Allergy, and Immunology, Department of Internal Medicine, Chiayi Chang Gung Memorial Hospital, Chiayi 613, Taiwan; 7Department of International Business, Ming Chuan University, Taipei 111, Taiwan

**Keywords:** competency-based medical education, CBME, competency, emergency physician, DEMATEL, ANP, VIKOR

## Abstract

The implementation of competency-based medical education (CBME) focuses on learners’ competency outcomes and performance during their training. Competencies should meet the local demands of the healthcare system and achieve the desired patient-centered outcomes. Continuous professional education for all physicians also emphasizes competency-based training to provide high-quality patient care. In the CBME assessment, trainees are evaluated on applying their knowledge and skills to unpredictable clinical situations. A priority of the training program is essential in building competency development. However, no research has focused on exploring strategies for physician competency development. In this study, we investigate the professional competency state, determine the driving force, and provide emergency physicians’ competency development strategies. We use the Decision Making Trial and Evaluation Laboratory (DEMATEL) method to identify the professional competency state and investigate the relationship among the aspects and criteria. Furthermore, the study uses the PCA (principal component analysis) method to reduce the number of components and then identify the weights of the aspects and components using the ANP (analytic network process) approach. Therefore, we can establish the prioritization of competency development of emergency physicians (EPs) with the VIKOR (Vlse kriterijumska Optimizacija I Kompromisno Resenje) approach. Our research demonstrates the priority of competency development of EPs is PL (professional literacy), CS (care services), PK (personal knowledge), and PS (professional skills). The dominant aspect is PL, and the aspect being dominated is PS. The PL affects CS, PK, and PS. Then, the CS affects PK and PS. Ultimately, the PK affects the PS. In conclusion, the strategies to improve the professional competency development of EPs should begin with the improvement from the aspect of PL. After PL, the following aspects that should be improved are CS, PK, and PS. Therefore, this study can help establish competency development strategies for different stakeholders and redefine emergency physicians’ competency to reach the desired CBME outcomes by improving advantages and disadvantages.

## 1. Introduction

A paradigm shift in medical education has been occurring in recent decades. The change is from the structure or process education style described by Abraham Flexner in 1910 to a competency-based education system [1,2]. An excellent description of competency-based medical education (CBME)—outlined by McGaghie and colleagues as early as 1978—emphasized that the outcome of CBME is a professional medical practice with a defined level of proficiency. The outcome of professional medical practice should also meet the needs of society and adapt to local conditions. The healthcare provider should have core professional competencies to satisfy the local community’s needs [3]. To implement the desired outcomes of CBME, we should build a competency framework. The most common examples are the six core competencies of the Accreditation Council for Graduate Medical Education (ACGME) and the seven roles of physicians in the CanMEDS framework [4,5]. Each competency framework consists of the defined and desired competencies for a physician. They are used to assess a physician’s ability to administer a high level of care to diagnose and treat illness, offer strategies to improve patient health, provide resources to prevent disease, and offer emotional support for patients and their families.

Although the competency framework has been developed for many years, it is challenging to measure the physicians’ abilities of competencies in practice [4,5,6]. Some studies applied descriptive narratives for the six core competencies to facilitate the outcomes evaluation. The ACGME and American Board of Emergency Medicine released the emergency medicine (EM) milestones as a framework for training programs to guide the assessment of trainees’ progress toward competence in each domain [7]. There are 23 “sub-competencies” within the EM milestones, each residing within one of the original six core competencies. Each sub-competency describes the knowledge, skills, attitudes, and behaviors (KSABs) for each level, ranging from novice to expert provider. Each description presents a particular developmental level of the trainee’s performance, and it’s an individual milestone. Individual milestones describe the KSABs required to progress from novice to expert [7].

Taiwan Society of Emergency Medicine (TSEM) is a professional organization of emergency medical physicians in Taiwan. TSEM provided continuous professional training and education to emergency physicians and residents to improve the quality of emergency care. In Taiwan, CBME has been implemented in some medical specialties, and emergency medicine is one of them. To implement the CBME and ACGME’s six core competencies in the clinical workplace, TSEM launched the Taiwan Milestone Project of Emergency Medicine in 2016 [8]. Twenty-three subcompetencies and 225 milestones were formulated by localization of the EM Milestone Project of ABEM and ACGME [9]. The project provided the institutions with a framework for residents’ training and education.

Since 2020, COVID-19 has impacted all healthcare professionals in every aspect of life, including emergency physicians (EPs), the first line of medical staff. They are affected by challenges, including disruption of patient care, work–life integration, mental health, and medical education [10,11,12,13]. To cope the threat of COVID-19, the type and strength of competency EPs required will be much higher than before. Based on the threat and disrupted training caused by the COVID-19 pandemic, we hope to provide effective development strategies to improve EPs’ competencies when facing the pandemic [14].

Previous studies on the competency of emergency physicians were focused on performance assessment tools, training, and improvement of a single subcompetency, and comparison of different frameworks and improvement [15,16,17,18,19]. There are few systemic and organized strategies to develop required competencies with priority. In this study, we first evaluate the causality and prioritization of EPs’ core competencies and analyze the differences in awareness and performance from various EPs. Additionally, we will then integrate these results to build development strategies and improvement paths for EPs’ core competencies.

There are five sections in the manuscript. Section 1 is the introduction which presents the background and motivation of the research. Section 2 explores the EPs’ requirement of abilities in the clinical workplace and constructs the evaluation model of professional core competency. Section 3 identifies the crucial influence element for healthcare using the hybrid Multiple Criteria Decision Making (MCDM) techniques. Section 4 analyzes the gap in competency development for the EPs, evaluates the advantage of professional competency, and identifies the strategy adoption paths of the stakeholders. Section 5 summarizes the findings and provides recommendations in conclusion.

## 2. Materials and Methods

This study was conducted in the emergency department (ED) of the Chang Gung Memorial Hospital Kaohsiung, Chiayi, and Linkou branches in Taiwan. All of these branches are teaching hospitals. Kaohsiung and Linkou branches are medical centers, and the Chiayi branch belongs to a regional teaching hospital. The study was approved by the Institutional Review Board (IRB) of the Chang Gung Memorial Hospital (IRB No: 202200680B0). We evaluated stakeholders (attending physicians, residents, and PGYs) at Chang Gung Memorial Hospital regarding professional competence development for EPs. There are four aspects and sixteen criteria introduced in the questionnaire. The four aspects include professional knowledge, professional skills, professional literacy, and care services. The questionnaire consists of three parts, the first is the basic information of respondents, the second is the survey of the awareness and performance about their competency state, and the third is perceptive about the relationship of aspects. Through an online questionnaire, the researcher collects the stakeholders’ awareness, performance, and the relationship of aspects on the questionnaire. An 11-point Likert Scale ranging from zero to 10 is used to gather the stakeholders’ views on the awareness and performance of the driving aspects/criteria. The survey was conducted from July 2022 to September 2022. The questionnaires did not include private data that would allow the identification of the participants. We informed all participants of the voluntary and anonymous nature of the study and documented their informed consent before the examination.

The MCDM was utilized to investigate the performance of various emergency physicians and develop the suited development paths. The hybrid approach of the MCDM method consists of seven steps: (1) defining critical decision problems, (2) developing the evaluation model with aspect/criteria, (3) building the network relation map with DEMATEL analysis, (4) identifying the critical component of criteria with PCA analysis, (5) using ANP analysis to measure the weights of the components of the structure, (6) ranking the alternatives with VIKOR analysis, and (7) establishing the suitable strategy adoption paths. The flowchart of the methodology is shown in Figure 1.

### 2.1. Establishing the Contents of Core Competencies for EPs

First, we conduct a literature review on the competency development of EPs and identify the critical aspects/criteria of competency development. We reviewed the core competencies of emergency physicians adopted by ABEM and ACGME [5,7]. Literature about competencies from other disciplines was also consulted [20,21,22]. Second, we set up a committee of experts for the competency development of EPs. The committee’s experts were asked for their cognitive debriefing about the aspects/criteria and questionnaire design to achieve face and content validity. There are seven experts on the committee, including three chiefs of the ED and four experienced expert emergency physicians at a different branch of Chang Gung Memorial Hospital. Experts in the committee were invited to confirm whether the critical aspects/criteria were appropriate or missing in the third step. According to the committee’s consensus and the literature review, we established the evaluation model of competency development for EPs. The evaluation model has four aspects: professional knowledge (PK), professional skills (PS), professional literacy (PL), and care services (CS). Each aspect consists of four evaluation criteria. The four aspects, sixteen related criteria, and their individual descriptions of the evaluation model are presented in Table 1.

#### 2.1.1. The Aspect of Professional Knowledge (PK)

Emergency medicine (EM) is a medical specialty that equips doctors with the knowledge and skills required to care for people with life-threatening or urgent healthcare needs. The ACGME and ABMS identified six core competencies required of residents and physicians to deliver high-quality medical care. In line with ACGME’s six core competencies, EPs require professional knowledge to perform patient evaluations and provide good healthcare. The majority of emergency patients lack diagnosis-specific symptoms [23]. The nonspecific complaint is the most common in the emergency setting, especially among the elderly. Patients with nonspecific complaints have a higher risk of mortality and morbidity than patients with specific complaints, so their care in the ED requires more time and resources [24]. Patients do not walk in the door wearing a label that says, “pulmonary embolism,” instead, they say they have a “cough.” The broad differential diagnosis of emergency patient complaints creates significant challenges for EPs [26]. Therefore, the professional knowledge to recognize the pitfalls in patient evaluation and management is crucial for EPs.

The model of emergency medicine practice is a complex process that begins with collecting a focused history of relative positives and negatives, completing a comprehensive physical, generating a differential diagnosis, and ordering all the appropriate investigations. Following these processes, the next step is providing the best care in the ED and making the subsequence disposition. If extended observation for diagnostic evaluation, prolonged therapy for identified medical conditions, or other ongoing assessment/management is necessary, patients should be observed in the ED or reassessed until their illnesses are treated correctly. These critical processes reveal several steps: focused history-taking and physical examination, pharmacotherapy, diagnostic studies, differential diagnosis, observation care, and reassessment [19,25,26]. Associated with the aspect of professional knowledge (PK), there are four evaluation criteria: diagnostic studies (PK1), pharmacotherapy (PK2), differential diagnosis (PK3), and observational reassessment (PK4). Those competencies are essential for EPs to provide high-quality patient care.

#### 2.1.2. The Aspect of Professional Skills (PS)

The ED is the gate of the hospital. Many critically ill patients who need urgent care and resuscitation are sent to ED. Emergency physicians have a great mission to effectively stabilize these critically ill patients using their professional skills [27]. To stabilize their patients’ condition, EPs should make all efforts to promptly achieve good outcomes for critical patients. The EPs cannot complete adequate resuscitation without these lifesaving procedures. Therefore, procedure management is a vital core competency for EPs required to stabilize their patients. Residents within emergency medicine programs require training in procedural skills, and emergency medicine attendings should be assessed to prevent skill decay [28]. Except for patient stabilization and procedure management abilities, rapid clinical evaluation to make diagnostic and therapeutic decisions is critical to EPs. Immediate evaluation relies on proficient clinical reasoning, focused history-taking, and physical examination [29]. Before practicing clinical reasoning, EPs gather the necessary information to develop a patient-specific problem list, which allows them to establish their differential diagnosis. However, there is limited time for EPs to gather detailed histories from the patient and perform complete physical examinations. Sometimes the patients in ED are unable or only partially able to communicate, making it challenging to convey the results of the history and clinical examination to the EPs. The most critical factor in restricted information gathering is that the EPs should identify patients who are about to go into cardiac or respiratory arrest within a short time. They should immediately provide resuscitation to prevent this situation, so they may have no time to perform history taking and clinical examination. As a result, achieving a focused history-taking and physical examination is a cornerstone for EPs in their clinical setting [30].

In recent decades, EPs have emphasized the utility of point-of-care ultrasound (POCUS) for ED patient care. Emergency medicine is one specialty that benefits from integrating POCUS into their practice and education [32]. POCUS is a quick, focused, bedside ultrasound examination performed by primary physicians and aimed at the real-time evaluation and management of patients in the ED. As such, it works hand-in-hand with a patient’s history and physical examination to identify the presence or absence of particular pathology. It can also be used to perform procedures to assure patient safety. There are several advantages of integrated POCUS in daily practice in ED. The advantages include quick diagnosis, cost-effectiveness, and shortening ED length-of-stay. POCUS can also improve laboratory testing wait time and reduce formal radiologic investigations. More and more specialties use POCUS to assess different situations, including surgery, trauma, pregnancy, and gynecologic, pediatric, and internal medicine patients. Therefore, POCUS has been a core competency of emergency medicine training and is listed as a milestone assessment under the CBME framework. With the trend of POCUS being integrated into ED practice, not only should residents be assessed before graduation, but EM faculty need to attain competence and become credentialed in POCUS [31]. The PS aspect comprises four evaluation criteria: emergency stabilization (PS1), clinical assessment (PS2), procedure management (PS3), and ultrasound use (PS4).

#### 2.1.3. The Aspect of Professional Literacy (PL)

Knowledge, skills, abilities, and behaviors (KSABs) are the primary element of each competency. In contrast to professional knowledge and skills, EPs also require professional abilities to practice emergency medicine safely and independently. The EPs are expected to demonstrate professionalism to patients, families, and staff at all times. The definition of professionalism varies in the literature and lacks consensus among organizations. Typical professionalism includes aspects of excellence, honor/integrity, altruism, respect for others, responsibility and accountability, collaboration, and communication skills. Various instruments have recently been developed to assess medical professionalism in EM residents [35,36,37]. Among emergency physicians, the critical task is to provide successful medical practice and prevent unprofessional behaviors and malpractices with credentialing professionalism. Among the six general competencies endorsed by ACGME, interpersonal and communication skills and professionalism are tied to the first two competencies that exemplary EPs should possess [34]. The third is practice-based learning and improvement. Therefore, non-technical skills, including professionalism, interpersonal communication, and practice-based learning and improvement, play a crucial role in EPs’ professional literacy.

Interprofessional communication (IPC) is information sharing among healthcare workers to influence patient care. IPC emphasizes the role of effective communication and collaboration with parents, nurses, technicians, and administrative staff [52]. Lee et al. concluded that almost one-third of ED patients require consultation from specialist teams [53]. Effective telephone consultation skills aid in mitigating stress, conflict, and miscommunication, which is fundamental to the EPs. A pilot study developed a helpful method for teaching telephone communication skills to improve the core competency needed by EPs [54]. As documented in a systematic review article, the IPC training program should begin as early as in medical school and the junior residency stage [55]. Practice-based learning and improvement emphasize the development of evidence-based practice to improve patient care and participation in performance improvement to support ED operations. Maintaining EPs’ essential competencies is critical to delivering high-quality, evidence-based care. The maintenance of certification programs is one method to achieve this goal, but it requires passive learning strategies. Hence, simulation-based education is currently being implemented by practicing physicians to maintain their professional competency [33].

In clinical practice, EPs experience stress when facing undifferentiated acute symptoms and concurrent competing care demands. ED providers practice multitasking to maintain fast and efficient care in time-limited ED environments. Managing available ED team members and resources in real-time with efficient care for patients in the ED is a critical competency and professional literacy for EPs. Multitasking is often unavoidable in an ED setting. Although multitasking increases stress experiences for EPs, multitasking also facilitates professionals’ experiences of situation awareness [33]. Associated with the aspect of professional literacy (PL), there are four evaluation criteria: task transition (PL1), professional ethics (PL2), interpersonal communication (PL3), and performance improvement (PL4).

#### 2.1.4. The Aspect of Care Services (CS)

ACGME and ABEM endorsed the emergency medicine milestones as a framework for developing and accessing trainees’ progress toward competence in each domain in 2012 [56,57]. Milestones describe competencies and identify KSABs that can be used as outcome measures within the six core competencies. Individual milestones demonstrate the KSABs required for graduation via unsupervised practice. However, some behaviors are not fully encompassed within KSABs in different social cultures and are categorized as care services (CS). With the aspect of care services (CS), there are four evaluation criteria: teamwork (CS1), patient safety (CS2), system management (CS3), and technology applications (CS4).

Many studies investigate the relationship between teamwork and patient safety and find a positive impact on it [38] in ED. Teamwork is a competency integrated with common thoughts, behaviors, and feelings that help health providers work as a team to provide better patient safety and outcomes in the clinic [38,42]. EM is considered a high-risk specialty in which inter-professional healthcare workers must engage to guarantee patient safety. Studies have shown that successful teamwork and communication training has improved patient outcomes [38,39,40]. However, Aouicha et al. described a worrying situation among those healthcare professionals in ED who lack a patient safety culture in their practice [41]. Therefore, teamwork training and building facilitate the cultivation of a patient safety culture and subsequently affect patient outcomes positively. The hospital can consider implementing effective teamwork and patient safety culture programs to reduce the incidence of unsafe care and adverse events.

System management focuses on both SBP and transition of care. SBP is the ACGME core competency that focuses on complex systems and physicians’ roles. SBP encompasses several topics, including multidisciplinary team-based care, healthcare quality improvements, cost containment, value consideration, and benefit/risk analysis to patient care. The definition of SBP does not easily translate into clinical observations and behaviors to assess in clinical practice. Gonzalo et al. developed five domains to evolve further SBP: comprehensive systems-based learning, the continuum of professional development, teaching and assessment methods, clinical learning environments of SBP, and professional identity [43]. A consensus conference on education research by Academic Emergency Medicine reviewed the literature on SBP assessment tools. It suggested multimodal assessment with direct observation by expert clinicians in the workplace [46]. In brief, EPs not only deliver effective, efficient, safe, timely, and patient-centered care, but also develop strategies to improve healthcare delivery within the ED, hospital system, and community. Care transitions occur when one healthcare provider transfers responsibility for a patient’s care to another. The evolution of patient care may happen between pre-hospital and ED providers, EPs at shift change, EPs and hospitalists, and ED and nursing homes [44,45,47]. All types of healthcare workers participate in the transition of a patient’s care. ED is considered a high-risk, unpredictable, and frequently interrupted environment, which may adversely impact patient care quality. The transition of care from the ED has significant risks for EPs and patients, and the competency of providing high-quality care is crucial to EPs. Unlike primary care, Rider et al. highlighted the importance of optimizing technology for an effective transition of care from the ED to the outpatient clinic [49].

The use of technology applications has increased in recent years, especially in the ED specialty, due to the utilization of artificial intelligence (AI). AI is considered the next major technological breakthrough in the healthcare system. EM has been at the forefront of disciplines using AI applications for patient care because of the uniqueness of the EM model. AI was adopted for clinical practice in numerous applications within EM, including in the interpretation of diagnostic imaging, interpretation of electrocardiography, and outcome prediction [51]. The intervention of AI can increase the speed and accuracy of clinical decisions and pose benefits to both EPs, ED, and healthcare systems [50]. The most established applications of AI in EM are within the ED itself. For example, AI has shown promise in interpreting diagnostic imaging, predicting patient outcomes, and monitoring patient vitals. However, facing the new technologies, EPs require careful vetting, legal regulations, patient safeguards, and user education. EPs should identify the limits and risks of AI while enjoying its potential benefits [48].

### 2.2. DEMATEL

The DEMATEL method is adapted to solve complex issues such as project selection, product planning, decision-making analysis, and strategy evaluation [58,59,60,61,62,63,64,65,66,67,68,69,70]. It is often applied in management, business, environment, and technology. In the medical research field, DEMATEL has been widely used in recent years, such as in evaluating the relationship between COVID-19 and acute kidney risk factors in elderly patients with DEMATEL [71], using the MCDM method to develop nosocomial infection control strategies [72], identifying the roles of social medial and emergency preparedness during the COVID-19 pandemic with fuzzy DEMATEL and ANP approach [73], and establishing the suited adoption paths and development strategy of the shared decision-making competencies with the DEMATEL technique [74]. This DEMATEL technique encompasses the following five steps: the first is finding the original average matrix; the second is estimating the direct influence matrix; the third is generating the indirect influence matrix; the fourth is evaluating the total influence matrix; and the fifth is obtaining the network relation map (NRM). The detailed calculated process and equations are in the Appendix A.

### 2.3. PCA (Principal Component Analysis)

The study analyzes the primary information on performance levels with the principal component analysis (PCA). The PCA approach helps to diminish the number of criteria without any information loss and aids the result analysis [75]. PCA is an essential component for analytic network procedure analysis. The detailed calculated process and equations are in the Appendix A.

### 2.4. Analytic Network Procedure (ANP)

To solve the various decision elements of interdependence and interrelationships in the world, the ANP can provide an effective solution [73,76,77,78,79]. The ANP technique can consider all the interactions and relationships among decision-making levels with a comprehensive hierarchical framework [80,81,82]. The ANP technique includes six steps as follows: (1) determine the research problem and build the evaluation of the framework, (2) design the evaluation framework and questionnaires investigation, (3) perform the pairwise comparisons matrices to determine the aspects/components’ weights with thinking about the feedback and dependence, (4) calculate the transposed matrix and derive the normalized matrix, (5) calculate the weighted supermatrix, and (6) compute the component’s weights [83,84,85]. After these serial steps, the relative weights of all the aspects and criteria in the evaluation model can be calculated with the ANP analysis. Finally, the weights will be utilized by VIKOR analysis to rank the alternatives. The detailed calculated process and equations are in the Appendix A.

### 2.5. VIKOR

Based on the VIKOR approach, we can obtain the positive and negative ideal solutions for the existing competency of emergency physicians. The scores of each criterion need to be summarized when calculating the gap between the competency status and the ideal solution. The gaps between the EPs’ most excellent awareness and the slightest awareness of the existing competencies can help to analyze the state of EPs’ current competency. Therefore, the researcher can use the VIKOR analysis to evaluate the CDI (competency development indicators) for emergency physicians and improve it [86,87,88,89,90,91,92,93,94,95]. The VIKOR approach uses an aggregate function in a compromise analytic process at the beginning [96,97]. It provided a maximum group utility of the “majority” and a minimum individual regret of the “opponent.” Compromising solutions can solve a discrete decision-making problem with non-commensurable and conflicting criteria. The decision-maker can apply the aspects/components’ weights to dialog, negotiate, and propose a compromise solution, as illustrated in Appendix A. The detailed calculated process and equations are also in the Appendix A.

## 3. Results

### 3.1. The Demographic Profile of the Valid Samples

Three teaching hospitals collected 135 questionnaires, and 117 were valid samples. There were 85 (72.6%) males and 32 (27.4%) females, as shown in Table 2, and 44 (37.6%) of the respondents were PGYs, 36 (30.8%) were emergency residents, and 37 (31.6%) were visiting staff. Most of these respondents are 26–30 years old and account for over 50%. They all came from Chang Gung Memorial Hospital, 76.9% belonged to the medical center branch, and the others were from the regional branch.

### 3.2. The Network Relation Map (NRM)

With the DEMATEL approach, we obtained the NRM (network relation map), as shown in Figure 2. The influencing aspects are PK (professional knowledge), PL (professional literacy), and CS (care services), while the affected aspect is PS (professional skills). The DEMATEL-NRM can calculate the degree of influence of all aspects and obtain the net impact relationship between these four aspects. The PL aspect has a net influence on PS, PK, and CS. Besides, the CS aspect has a net influence on the PK and PS. The PK aspect has a net influence on PS. Therefore the PL aspect can enhance first, followed by the CS and PK aspects. The aspect that least needs improvement is the PS among the four aspects, as illustrated in Figure 2.

### 3.3. The Comparative Analysis of Competency Development for Emergency Physicians

#### 3.3.1. The Overall Views of Competency Development for Emergency Physicians

This study selects three different ranks of physicians in ED (attending physician/visiting staff, resident, and PGY) in Taiwan. The study constructs the four aspects (professional knowledge, professional skills, professional literacy, and care services) to evaluate emergency physicians’ CDI (competency development indicators). This study also compares the competitive dominance of the CDI for the three occupational classes of physicians in ED. In the horizontal analysis, the visiting staff (Staff C) have competitive dominance in the PK (professional knowledge), PL (professional literacy), and CS (care services) aspects, while the residents (Staff B) have a competitive dominance in the aspect of PS (professional skills). In contrast to PS, the residents have a competitive weakness in PK, PL, and CS aspects. The PGYs (Staff A) have a competitive weakness in the aspects of PS, as illustrated in Figure 3 and Table 3. In the vertical analysis, the PGYs, the residents, and the visiting staff all have a weakness/gap in the CS aspect. The PGYs and the visiting staff have competitiveness in the PK aspect. The residents have competitiveness in the PS aspect, as shown in Figure 3 and Table 3.

#### 3.3.2. The PK (Professional Knowledge) Aspect of Competency Development for Emergency Physicians

In the study, there are four criteria within the PK aspect (diagnostic studies, pharmacotherapy, differential diagnosis, and observational reassessment) to evaluate the CDI for the three different ranks of physicians in ED. In the horizontal analysis, the PGYs (Staff A) have competitive dominance in the criteria of PK1 (differential diagnosis) and PK2 (pharmacotherapy), and the visiting staff (Staff C) have competitive dominance in the criteria of PK3 (differential diagnosis) and PK4 (observational reassessment). The residents (Staff B) have a competitive weakness in all criteria (PK1, PK2, PK3, and PK4), as illustrated in Figure 4 and Table 4.

In the vertical analysis, the PGYs and the residents have the advantage/competitiveness in the criterion of PK1. The PGYs and the residents have a weakness in the criterion of PK4, as shown in Figure 4 and Table 4. The visiting staff have a weakness/gap in the criterion of PK2. The visiting staff have a competitive advantage in the criterion of PK3. The PGYs and the residents have a weakness in the criterion of PK4, as shown in Figure 4 and Table 4.

#### 3.3.3. The PS (Professional Skills) Aspect of Competency Development for Emergency Physicians

In the study, there are four criteria within the PS aspect (emergency stabilization, clinical assessment, procedure management, and ultrasound use) to evaluate the CDI for the three different ranks of physicians in ED. In the horizontal analysis, the PGYs (Staff A) have the competitive advantage in the criteria of PS2 (clinical assessment), and the residents (Staff B) have the competitive dominance in the criteria of PS3 (procedure management) and PS4 (ultrasound use). The visiting staff (Staff C) have a competitive dominance in the criteria of PS1 (emergency stabilization). The PGYs have a competitive weakness in the criteria of PS1 and PS3, whereas the residents have a competitive weakness in the criteria of PS2, and finally, the visiting staff have a competitive weakness in the criteria of PS4.

In the vertical analysis, Staff B and Staff C are competitive in the PS1 aspect, and Staff A is competitive in the PS2 aspect. Staff A, B, and C all have competitive weaknesses in the criteria of PK4, as shown in Figure 5 and Table 5.

#### 3.3.4. The PL (professional literacy) Aspect of Competency Development for Emergency Physicians

In the study, there are four criteria within the PL aspect (task transition, professional ethics, interpersonal communication, and performance improvement) to evaluate the CDI for three different occupational classes of physicians in ED. In the horizontal analysis, the visiting staff (Staff C) have a competitive dominance in the criteria of PL1 (task transition), PL2 (professional ethics), PL3 (interpersonal communication), and PL4 (performance improvement). The PGYs (Staff A) have a competitive weakness in PL1, and the residents (Staff B) have a competitive weakness in the PL2, PL3, and PL4 criteria, as shown in Figure 6 and Table 6.

In the vertical analysis, the PGYs are competitive in the criteria of PL2, and the residents are competitive in the PL4 criteria. The visiting staff have competitiveness in the criteria of PL3. Staffs A and B have a disadvantage in the criteria of PL1, and Staff C have a weakness in the criteria of PL1 and PL4, as shown in Figure 6 and Table 6.

#### 3.3.5. The CS (Care Services) Aspect of Competency Development for Emergency Physicians

In the study, there are four criteria within the CS aspect (teamwork, patient safety, system management, and technology applications) to evaluate the CDI for the three different ranks of physicians in the ED. In the horizontal analysis, the PGYs (Staff A) have competitive dominance in the criteria of CS2 (patient safety) and CS3 (system management), and the visiting staff (Staff C) have competitive dominance in the criteria of CS1 (teamwork) and CS4 (technology applications). The residents (Staff B) have a competitive weakness in the CS1, CS2, CS3, and CS4 criteria, as illustrated in Figure 7 and Table 7. In the vertical analysis, Staffs A, B, and C dominate the criteria of CS2 and have weaknesses in the criteria of CS4, as in Figure 7 and Table 7.

### 3.4. PCA (Principal Component Analysis)

The number of components can be introduced with the PCA approach. If the eigenvalue is larger than one (*λ_t_* > 1), the possible component is reserved; on the contrary, the component is removed. Under the PCA approach, the aspect of professional knowledge (PK) accumulation loading is 79.993%, which can explain 79.993% of the aspect content. We can determine the accumulating loading threshold, such as 70%, and factor loading to determine the components’ number. Each CDI aspect can extract one principal component, as illustrated in Table 8. We can rename each component as follows respectively: diagnostic and therapeutic knowledge (PKP1), stabilization and procedural skills (PSP1), professionalism and interpersonal communication (PLP1), and system management and teamwork (CSP1), as illustrated in Table 8.

### 3.5. Analytic Network Procedure (ANP)

The calculated weights of aspects and components are shown in Table 9. Based on the PCA approach, the highest component weight is the PS aspect (0.259), followed by the aspect of PK (0.257), CS (0.246), and PL (0.238).

### 3.6. The Ranking of Professional Competency for Emergency Physicians under the VIKOR Approach

The VIKOR method is a multi-criteria decision-making process. It can help decision-makers to rank and select from a set of alternatives and finally produce a final solution. Therefore, the CDI (competency development indicators) of different alternatives can be acquired by the VIKOR analysis, as shown in Table 10.

As illustrated in Table 10, under v = 0.5, the lowest CDI is 0.775 belonging to Staff B (the residents), and the highest CDI is 0.815 belonging to Staff C (the visiting staff). The ranking of the CDI of the emergency physicians is Staff C⊃Staff A⊃Staff B. The visiting staff is higher than the PGYs and the residents. Under v = 1.0, the lowest CDI is 0.796 and belongs to Staff B; the highest CDI is 0.829 belonging to Staff C; the ranking of the CDI of emergency physicians is Staff C⊃Staff A⊃Staff B. The emergency physicians of Staff C are better than other levels of seniority, as shown in Table 10.

The decision-makers can apply the ranking under v = 0 or v = 1.0 to evaluate the worst and best conditions. The ranking of CDI can be assessed under stable conditions so that we use the suitable CDI for the emergency physicians v = 0.5, as illustrated in Table 10. Under different v values in the study, our analysis has the same CDI ranking result. Emergency physicians can use powerful analytics to understand the CDI rank in the three states and determine the improving routes for competency development. By the ranking of CDI, the visiting staff have the highest value in CDI, and the residents have the lowest value in CDI. Therefore, the residents should be enhanced first with the four driving forces to improve their competencies, as illustrated in Figure 3 and Table 3.

### 3.7. The Improving Paths Analysis for Different Stakeholders of Emergency Physicians

This study analyzes the aspect performance to introduce EPs’ strategy adoption paths of competency. The dominant aspect can improve the weak aspect so that higher performance can enhance lower performance. In Staff A (The PGYS), the EPs’ aspect performance rank was PK (professional knowledge) > PS (professional skills) > PL (professional literacy) > CS (care services), as illustrated in Table 11. Then, the EPs of Staff A can determine four available paths, and three of them are strategy adoption paths [PL(3)PK(1)→PS(2); PL(3)→CS(4)→PS(2); PL(3)→PS(4)→PK(1)→PS(2)]. In Staff B (the residents), the rank of aspect performance for the EPs was professional skills (PS) > professional knowledge (PK) > professional literacy (PL) > care services (CS), as illustrated in Table 11. Then, the EPs of Staff B can determine four available paths, and two of them are strategy adoption paths [(PL(3)→CS(4)→PS(1); PL(3)→CS(4)→PK(2)→PS(1)]. In Staff C (the visiting staff), the rank of aspect performance for the EPs was professional knowledge (PK) > professional literacy (PL) > professional skills (PS) > care services (CS), as illustrated in Table 11. Then, the EPs of Staff C can determine four available paths, and all of them are strategy adoption paths [(PL(2)→PS(3); PL(2)→PK(1)→PS(3)); PL(2)→CS(4)→PS(3); PL(2)→CS(4)→PK(1)→PS(3)]. Thus, the EPs can determine two common strategy adoption paths (PL→CS→PS; PL→CS→PK→PS) as illustrated in Table 11.

Therefore, the decision maker can analyze the aspects/components’ rank (advantageous/disadvantageous) as illustrated in Table 11 and employ the strategy adoption paths to enhance the competency development of EPs.

## 4. Discussion

### 4.1. Identifying the Gap in the Competency Development of EPs

CBME is gaining momentum in the field of medical education across the world. It emphasizes outcome-based training and promises greater accountability, flexibility, and learner-centeredness [98]. To be competent, health professionals should have the necessary ability, knowledge, and skills to do something successfully and efficiently. The required ability, knowledge, and skills are called “competency” [99]. The core competencies required of a medical graduate physician are endorsed by ACGME in the USA, GMC in the UK [100], and CanMEDS in Canada. Based on the core competencies, many studies focus on assessing the competencies, designing the training curriculum, and integrating the competencies into the clinical workplace. Assessment tools are developed to evaluate the fitness of the desired competencies, including the six core competencies. According to the results of assessments, the training program can adjust its training and education and provide tailored learning and improvement. Everyone knows what competencies a physician should have and what tools to evaluate them.

Still, more is needed to study whether there is a recommended training sequence in the competency development process. We searched the literature for relevant information to confirm the order of competency development; however, we have not found any studies focused on exploring the strategies for physician competency development. Of course, the ACGME addressed milestone 2.0 to EPs and guided their competency development for the future [101]. Modified subcompetencies are declared for EPs’ training, and each subcompetency is equally essential. The competencies are often divided into three domains, knowledge, skills, and attitude, so that the institution can identify the competency gap from each domain [102]. Improvement strategies are focused on a single domain or subcompetency, for example, interprofessional communication skills, ultrasound training, patient safety, and care transitions [18,47,55,103]. In our research, we first confirm what abilities emergency physicians should have, analyze the current situation of emergency physicians’ abilities, and then develop the order of competency development of physicians of different ranks. In general, competencies include knowledge, skills, attitudes, and behaviors. Is competency development a priority of knowledge or skills? Or does attitude matter above all else? In 1973, David McClelland argued intelligence alone was not an indicator of success or failure in the workplace [104]. Therefore, the orderly training model can promote the balanced development of professional competencies. Determining the priority for competency development is a critical issue and choice. According to our study results, we develop the strategic priority for three different ranks of EPs.

### 4.2. Identifying Strategy Adoption Paths of the PGYs (Staff A)

Staff A have three strategy adoption paths (PL→PK→PS; PL→CS→PS; PL→CS→PK→PS) based on the NRM (network relation map) approach and the CDI (competency development indicators) rank, as illustrated in Table 11. The PGYs had an advantage in the PK (professional knowledge) and PS (professional skills) aspects but a disadvantage in the PL (professional literacy) and CS (care services) aspects. The PGYs will be responsible for clinical practice and patient care during their ED rotation. They are unfamiliar with the ED and the healthcare providers in ED, so the PGYs lack the ability for teamwork management. It is harmful to teamwork development in ED practice. They also cannot provide optimal healthcare in the system due to their limited ED rotation. Therefore, teamwork and system-based management are the primary competencies that should improve for PGYs. In Staff A, the second strategy adoption path (PL→PK→PS), the third strategy adoption path (PL→CS→PS), and the fourth strategy adoption paths (PL→CS→PK→PS) can apply to Staff A EPs to improve their competency development. The PK can strengthen the PS in the second strategy adoption path. For example, the Staff A EPs can increase their understanding of pharmacotherapy by enhancing self-learning and performance improvement. Yet, the EPs’ strength in the knowledge of diagnostic studies also improves clinical assessment. The PL can enhance the CS in the third strategy adoption path. The Staff A EPs can strengthen teamwork and collaboration through professionalism, and excellent teamwork enhances the efficiency of procedure management. The PL can improve the CS, and then the PK improves the PS in the fourth strategy adoption path. The Staff A EPs can enhance their interpersonal communication to strengthen the quality of patient safety and system management. Then, the EPs with the knowledge of diagnostic studies and differential diagnosis can improve their emergency stabilization skills.

### 4.3. Identifying Strategy Adoption Paths of the Residents (Staff B)

The Staff B EPs had two suited improvement routes (PL→CS→PS; PL→CS→PK→PS) based on the CDI rank, as shown in Table 11. The residents had an advantage in the PK (professional knowledge) and PS (professional skills) aspects but a disadvantage in the PL (professional literacy) and CS (care services) aspects. In Staff B, the third strategy adoption path (PL→CS→PS) and fourth strategy adoption path (PL→CS→PK→PS) can apply to Staff B EPs. The residents play a significant role in clinical practice and are more aware of health system resource coordination and improvement. They have the sense to keep improving and promote self-learning. Therefore, the CS is a weak competency possessed by the residents in the study. According to the adoption path, the PL can improve the CS in the third strategy adoption path. For example, the Staff B EPs can strengthen their teamwork with other health professionals through training in professionalism and interpersonal communication skills. The PL can improve the CS in the fourth strategy adoption path. The residents can optimize the ED function by self-learning and performance improvement, then enhance patient care quality and prompt the strategies developed to provide optimal healthcare. The improved competency of CS can provide better diagnostic tools and the ability to reassessment. The PS can be improved after strengthening the PK. Therefore, the residents can enhance their clinical assessment and procedure management competencies.

### 4.4. Identifying Strategy Adoption Paths of the Visiting Staff (Staff C)

The Staff C EPs had four strategy adoption paths (PL→PS; PL→PK→PS; PL→CS→PS; PL→CS→PK→PS) based on the CDI rank, as shown in Table 11. The visiting staff had an advantage in the PK (professional knowledge) and PL (professional literacy) aspects but a disadvantage in the PS (professional skills) and CS (care services) aspects. The visiting staff’s engagement directly links to the quality of patient care, with exemplary professionalism and individual performance improvement recognized as necessary to achieve excellence in care services. These services—including maintaining patient safety, promoting teamwork, and applying system-based management—can establish a safe ED workplace and medical supply with a high level of proficiency. In Staff C, the first strategy adoption path (PL→PS), the second strategy adoption path (PL→PK→PS), the third strategy adoption path (PL→CS→PS), and the fourth improvement route (PL→CS→PK→PS) all can apply in Staff C EPs. The PL can improve the PS on the first strategy adoption path. For example, patient safety can improve by enhancing the professionalism of the visiting staff. PK can improve PS in the second strategy adoption path. The Staff C EPs can strengthen diagnostic studies by enhancing performance improvement.

Furthermore, the visiting staff strengthens the ultrasound skill by increasing the knowledge of the differential diagnosis. The PL can enhance the CS in the third strategy adoption path. The Staff C EPs can strengthen patient safety by applying interpersonal communication and enhancing clinical assessment. The PL can improve the CS, and then the PK can enhance the PS in the fourth strategy adoption path. The EPs can enhance patient safety and system management by strengthening their professionalism and increase observational reassessment by strengthening the technology applications. Besides, the EPs can increase their knowledge of differential diagnosis to enhance their procedure management and clinical assessment skills.

### 4.5. Identifying Common Strategy Adoption Paths for the Three Stakeholders of EPs

The study combined the strategy adoption paths of the three stakeholders of EPs and introduced the common strategy adoption paths. In the view of Staff A (the PGYs), the strategy adoption paths are PL→PK→PS, PL→CS→PS, and PL→CS→PK→PS. In the view of Staff B (the residents), the strategy adoption paths are PL→CS→PS and PL→CS→PK→PS. In the view of Staff C (the visiting staff), the strategy adoption paths are PL→PS, PL→PK→PS, PL→CS→PS, and PL→CS→PK→PS. There are two introduced common strategy adoption paths, which are PL→CS→PS and PL→CS→PK→PS, as shown in Table 11. Emergency physicians can adopt the priority of development paths to enhance the competencies required in the clinical workplace. By strengthening the aspect of PL, the CS, PK, and PS aspects can be improved sequentially. The PL aspect includes four evaluation criteria: task transition (PL1), professional ethics (PL2), interpersonal communication (PL3), and performance improvement (PL4). Therefore, to enhance the competency development of various EPs in the ED, the administrators should emphasize training in interpersonal communication, professionalism, self-performance improvement, and the ability to task transition, not paying attention to medical knowledge or procedural skills [105,106,107].

### 4.6. Ranking of Professional Competency for Emergency Physicians Based on the VIKOR Approach

When ranking the professional competency of EPs under the best or worst conditions, the visiting staff have the best performance and the residents have the worst expression, as shown in Table 10. According to the result of the VIKOR analysis, the resident is the stakeholder who should be first improved and enhanced. To provide effective and efficient patient care in ED, the visiting staff are the best, and the PGYs are the next. In Taiwan, the PGYs should visit ED with two months of rotation training. The PGYs can provide primary care and medical assistance, such as simple procedural management, patient transfer, and system-based practice. The residents can focus on high-risk procedural management and patient care. Based on the convenience of medical support by the PGYS, the residents’ competencies seem to degenerate except for professional skills. There is no literature comparing PGYS and ED, but there is literature comparing rural and urban EPs [108]. The weakening of residents’ competencies is a warning sign in the competency development of EPs. The study aids hospital administrators and training program directors in reevaluating the strategies of professional competency development for EPs.

### 4.7. Comparing our Study with Previous Studies

The literature regarding competency development focused on competency evaluation and assessment, barriers to competency development, maintenance of competency, competency model building, and a novel competency-based medical education implementation [19,27,28,33,34,109,110]. There is little document to prioritize the strategies adaption paths about the professional competency development of EPs. As described by ACGME’s six core competencies and EM milestones, each milestone demonstrated knowledge, skills, and attitudes required by EPs to perform a successful task within a given situation. In the pedagogical trinity model, interrelated knowledge, skills, and attitudes will develop interdependently [111,112]. People can solve problems with the required competencies constructed with the trinity model [113]. Our research finds a common improvement strategy to enhance the EPs’ competencies. Through the improvement of PL, the CS, PK, and PS will then be improved. The aspect of PL comprises non-technical skills and focuses on the attitudes dimension. Therefore, knowledge and skills can be strengthened by improving the attitudes dimension. Our study results are coherent with some research. The attitudes dimension can affect the transformation of knowledge and skills, proposed by Cooley et al.’s research [114]. If learners enjoy the process of learning, a new context transformation of skills and knowledge will more likely occur. Grossman and Salas demonstrate that people with stronger motivation and self-efficacy will practice good transformation of skills in workplace training [115]. Overall, our study agrees with their viewpoint that attitudes can influence the transformation of knowledge and skills. Hence, facilitating the enhancement of PL is a preferred strategy to prompt the competency development of EPs.

The study uses the MCDM approach to investigate the competencies development state, establish the strategy adaption paths, and rank the alternatives. Therefore, based on each stakeholder’s competitive advantages of professional competency, our study results can aid the EPs in recognizing the competency state and constructing the improvement strategies for their competency development. Under the utilization of the NRM approach, the proposed CDI model can help EPs to identify crucial impacts and determine the appropriate improvement strategies.

## 5. Conclusions and Recommendations

### 5.1. Conclusions

To promote excellence, the GMC establishes the standard requirements for all graduated physicians in medical education and training [100]. There are three outcomes in the “outcomes for graduates.” The first is professional values and behaviors, the second is professional skills, and the third is professional knowledge. Highly competent professionals are critical to improving healthcare quality. It is compatible with the description of competency. Physicians of different ranks have their competitive advantages and disadvantages. The competitive advantages can improve weaknesses, and higher performance can enhance lower performance. The PGYs have three strategy adoption paths (PL→PK→PS; PL→CS→PS; PL→CS→PK→PS), the residents have two strategy adoption paths (PL→CS→PS; PL→CS→PK→PS), and the visiting staff have four strategy adoption paths (PL→CS; PL→PK→PS; PL→CS→PS; PL→CS→PK→PS). There are two common strategy adoption paths for the three different ranks of physicians (PL→CS→PS; PL→CS→PK→PS). Regardless of the rank of EPs and the strategy adoption path, to achieve the expected competency, we must strengthen professional literacy (PL) to improve the care services (CS). Therefore, EPs will naturally enhance their professional knowledge (PK) and professional skills (PS). This study provided an overview of the competency development strategy in which current and aspiring emergency physicians should proactively strengthen their excellence.

### 5.2. Academic Contributions

The structural equation modeling can evaluate the symmetry network relation structure, but it cannot identify the dominant relationship of network structure [116,117,118]. Therefore, four analytic methods integrate into our research model: the DEMATEL, PCA, ANP, and VIKOR approaches. The DEMATEL technique solves the limitation of structural equation modeling, and the DEMATEL technique evaluates the dominance relationship for network relation structure [66,69]. Emergency physicians can use the VIKOR approach to assess their CDI rank and determine professional competency benchmarks. They also use the strategy adoption path to enhance their professional competency development. This study introduces the methodology of DEMATEL and MCDM approaches to solve complex relationships and decision problems with conflicting and incommensurable criteria. The different improvement strategies of diverse physicians we offered in this study can provide critical information for competency development and valuable suggestions for various stakeholders.

### 5.3. Study Limitations

First, most questionnaires were obtained from the medical center (76.9%) and regional hospital (23.1%) in the research. The result of our study represents the competency development of EPs in the medical center. Because of the difference in community and background, the competencies required may differ in the medical center and regional hospital. Further study should focus on the difference in competency development of various levels of hospitals.

Second, the study proposes a model to modify the ANP through the PCA approach and uses the modified VIKOR approach to make the desired solution better. Therefore, the VIKOR approach still needs to be improved in determining the v value. This study compares the various v values and offers three different situations (v = 0, v = 0.5, and v = 1.0) for the robustness analysis of CDI. These researchers can also determine their suited v values for different states. The decision-makers should evaluate the condition for the decision problem. Then, decision-makers can adopt the CDI value under v = 0.0 in the pessimistic state and take the CDI value under v = 1.0 in the optimistic state. This study considers that decision-makers can evaluate CDI status in a stable and steady state. Therefore, this study finds the suitable CDI of emergency physicians can adopt the v = 0.5 in the general situation. However, the different condition has their reasonable v determination. Therefore, further research can consider various v determinations in multiple situations.

Third, this study is based on a self-administered questionnaire, not on a pre-post study, and does not show the effectiveness of the interventions. The proposed strategy is rooted in a cognitivistic theoretical framework that does not consider the role of social and educational interactions of each specific community of learners, which could have another preferred or more effective pathway.

### 5.4. Future Studies

Although the six general core competencies identified by ACGME are required for each physician, every specialty has its dominant competency and sub-competency. Competency development varied with different types of stakeholders, specialties, and hospitals. Further studies could compare the competency development between different types of stakeholders (nurses, pharmacists, radiologic technologists, etc.), specialties (surgery, internal medicine, pediatrics, obstetrics-gynecology, anesthesiology, ophthalmology, etc.), and hospitals (acute or long-term care, rural or urban, specialty hospitals, etc.).

## Figures and Tables

**Figure 1 healthcare-11-00471-f001:**
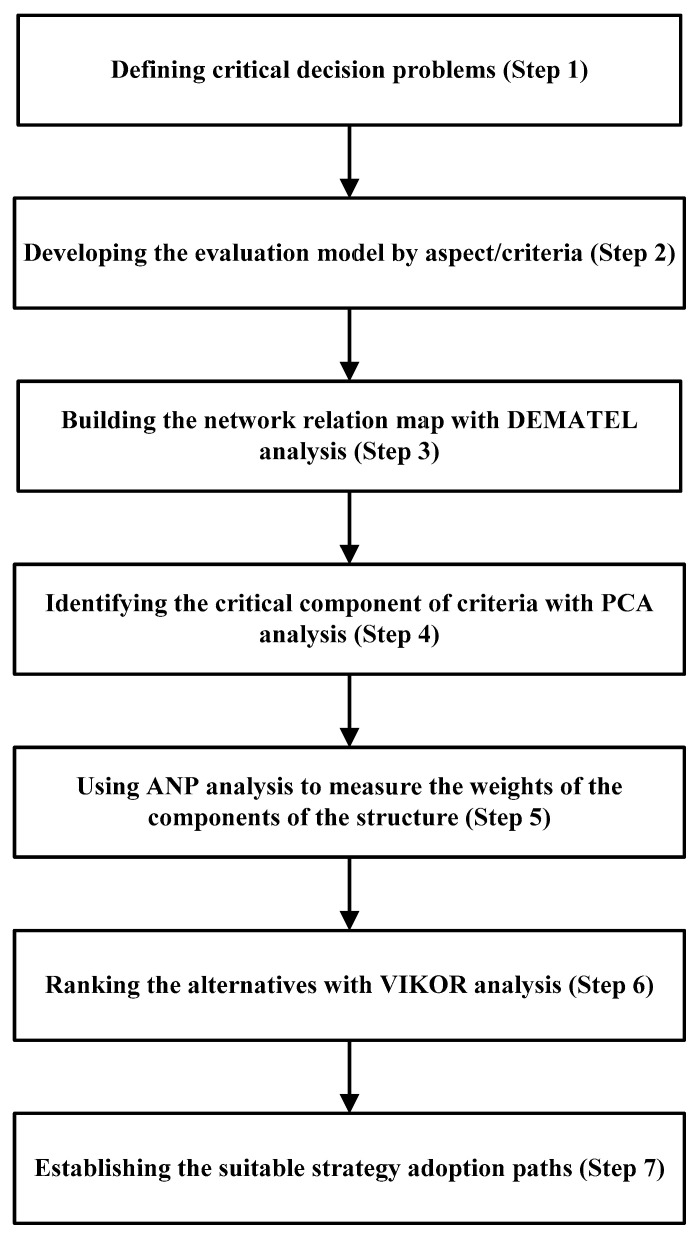
The Hybrid approach methods.

**Figure 2 healthcare-11-00471-f002:**
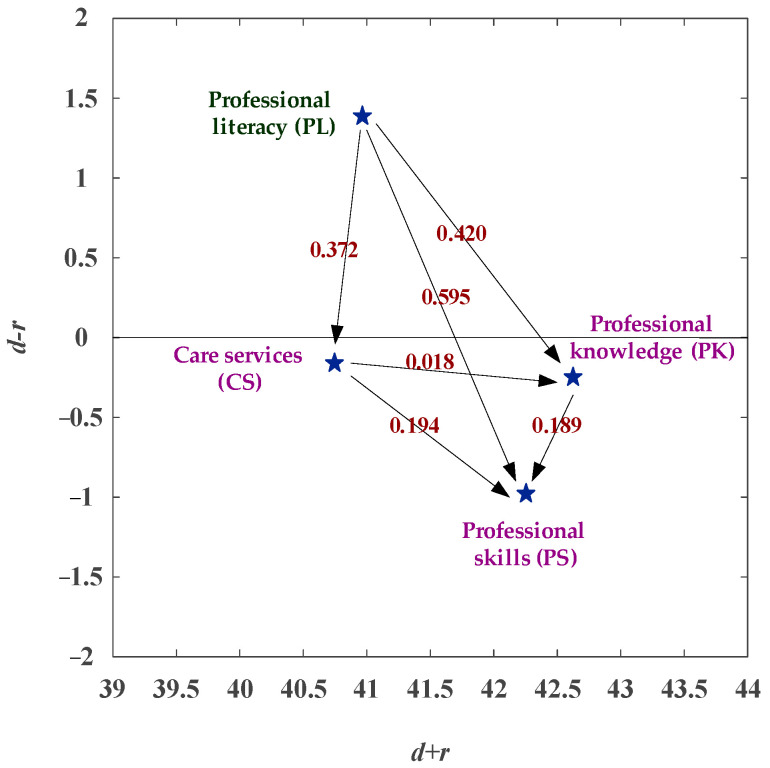
The NRM (network relation map) of competency development.

**Figure 3 healthcare-11-00471-f003:**
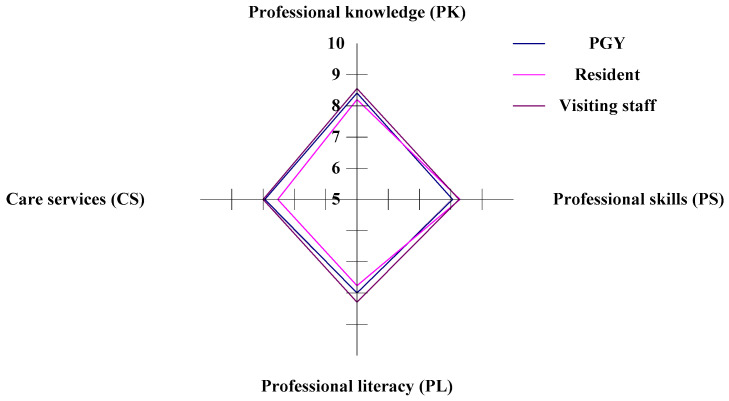
The radar chart of the competency development indicators (CDI).

**Figure 4 healthcare-11-00471-f004:**
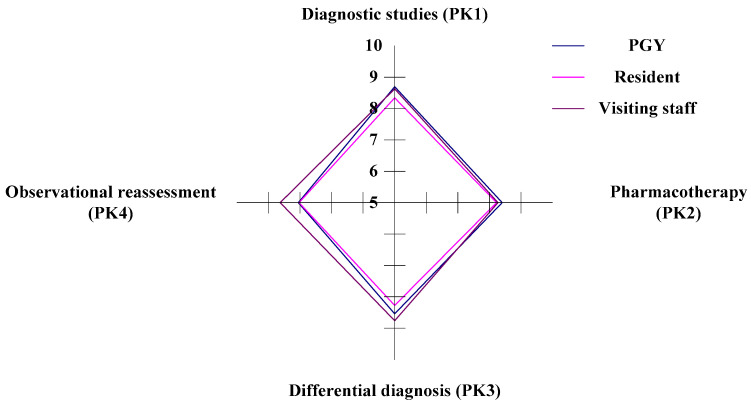
The situation of competency development in the PK aspect.

**Figure 5 healthcare-11-00471-f005:**
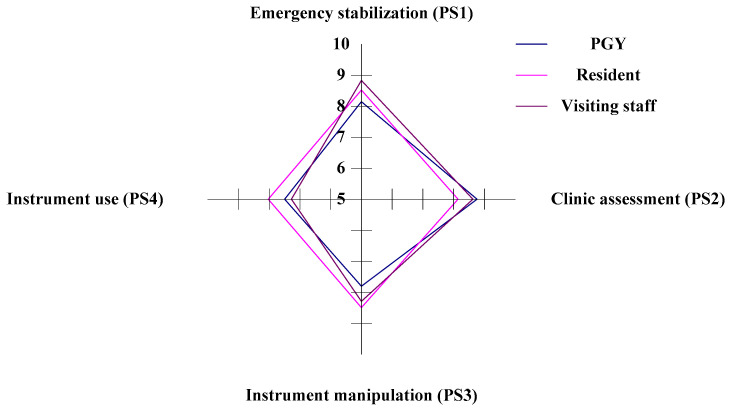
The situation of competency development in the PS aspect.

**Figure 6 healthcare-11-00471-f006:**
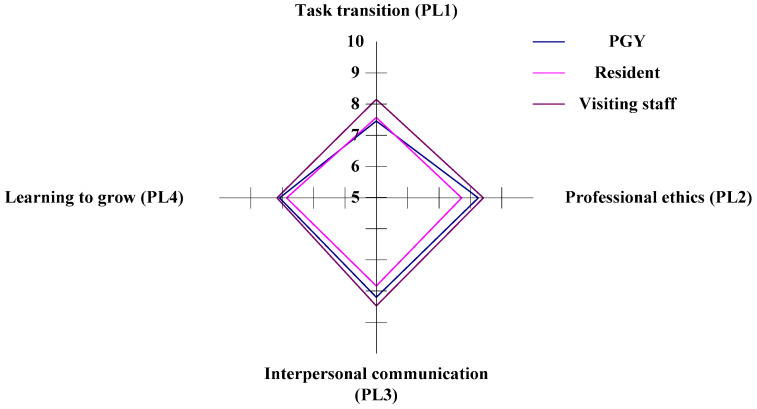
The situation of competency development in the PL aspect.

**Figure 7 healthcare-11-00471-f007:**
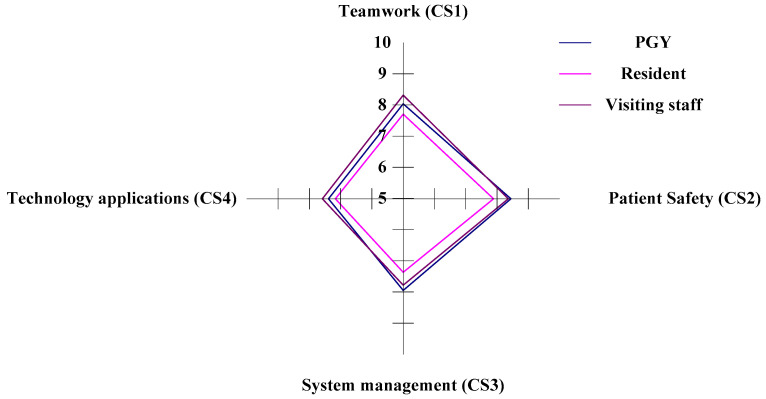
The situation of competency development in the CS aspect.

**Table 1 healthcare-11-00471-t001:** The descriptions of professional competency development for emergency physicians.

Aspects/Criteria	Descriptions	References
**Professional knowledge (PK)**		
Diagnostic studies (PK1)	Knowledge of diagnostic studies selection can help EPs improve their ability to diagnose and provide patient care.	[23,24]
Pharmacotherapy (PK2)	The development of pharmacotherapy knowledge can help EPs select appropriate agents for therapeutic intervention.	[19,25,26]
Differential diagnosis (PK3)	The development of differential diagnosis knowledge can help EPs prioritize the potential diagnosis based on the initial assessment.	[24,26]
Observational reassessment (PK4)	Developing knowledge of disposition decisions and patient education plans can improve patient safety and maximize resources.	[19,25,26]
**Professional skills (PS)**		
Emergency stabilization (PS1)	The development of emergency stabilization can help EPs to identify unstable patients and perform immediate interventions to stabilize patients before deterioration.	[27,28]
Clinical assessment (PS2)	Performing a focused history and physical exam can help EPs to effectively guide the diagnosis and management of the patient’s urgent issues and minimize the need for further diagnostic testing.	[29,30]
Procedure management (PS3)	Performing and interpreting procedures (e.g., endotracheal intubation, suturing, splinting, and vascular access) can help EPs to optimize aggregate patient outcomes within the system.	[27,28]
Ultrasound use (PS4)	Performing goal-directed focused ultrasound exams can help EPs to manage patient assessment and invasive procedures promptly.	[31,32]
**Professional literacy (PL)**		
Task transition (PL1)	The development of task transition can help EPs to manage available ED team members and resources efficiently under high volume or surge situations in the ED.	[33]
Professional ethics (PL2)	The development of professional ethics can ensure that EPs respect workplace ethical principles, demonstrate professional responsibilities, and optimize their personal and professional well-being.	[34,35,36,37]
Interpersonal communication (PL3)	Developing interpersonal communication strategies helps EPs mitigate stress, conflict, and miscommunication.	[28,29,30]
Performance improvement (PL4)	EPs apply evidence-based care to complex patients and commit to personal growth to facilitate individual and institutional improvement.	[33]
**Care services (CS)**		
Teamwork (CS1)	The development of teamwork management can facilitate EPs to resolve specific ED challenges through effective communication and mutual respect among team members.	[38,39,40]
Patient safety (CS2)	EPs engage the emergency department and hospital system to offer error-prevention strategies and prevent patient safety events.	[38,41,42]
System management (CS3)	EPs should develop and apply strategies to evaluate and improve healthcare supply to provide optimal healthcare in the system.	[43,44,45,46,47]
Technology applications (CS4)	EPs use electronic devices to provide efficient and effective medical practice, the transition of care, computerized processes, and learning.	[48,49,50,51]

**Table 2 healthcare-11-00471-t002:** The demographic profile of the valid samples (N = 117).

Characteristics		Number (%)
Gender	Male	85 (72.6)
	Female	32 (27.4)
Age	20~25	12 (10.3)
	26~30	59 (50.4)
	31~35	10 (8.5)
	36~40	9 (7.7)
	41~45	9 (7.7)
	46~50	13 (11.1)
	>51	5 (4.3)
Workplace institution	Medical center	90 (76.9)
	Regional hospital	27 (23.1)
Stakeholders	PGYs	44 (37.6)
	Residents	36 (30.8)
	Visiting staff	37 (31.6)

**Table 3 healthcare-11-00471-t003:** The situation of EPs’ competency development.

Aspects	PGYs	Residents	Visiting Staff
	(Staff A)	(Staff B)	(Staff C)
Horizontal analysis			
Professional knowledge (PK)	8.426 [2]	8.215 [3]	8.568 [1]
Professional skills (PS)	8.051 [3]	8.299 [1]	8.257 [2]
Professional literacy (PL)	8.000 [2]	7.750 [3]	8.297 [1]
Care services (CS)	7.955 [2]	7.535 [3]	8.014 [1]
Vertical analysis			
Professional knowledge (PK)	8.426 [1]	8.215 [2]	8.568 [1]
Professional skills (PS)	8.051 [2]	8.299 [1]	8.257 [3]
Professional literacy (PL)	8.000 [3]	7.750 [3]	8.297 [2]
Care services (CS)	7.955 [4]	7.535 [4]	8.014 [4]

**Table 4 healthcare-11-00471-t004:** The competency analysis of the PK aspect.

Staffs Criteria	PGYs	Residents	Visiting Staff
	(Staff A)	(Staff B)	(Staff C)
Horizontal analysis			
Diagnostic studies (PK1)	8.705 [1]	8.333 [3]	8.622 [2]
Pharmacotherapy (PK2)	8.409 [1]	8.222 [3]	8.270 [2]
Differential diagnosis (PK3)	8.545 [2]	8.278 [3]	8.757 [1]
Observational reassessment (PK4)	8.045 [2]	8.028 [3]	8.622 [1]
Vertical analysis			
Diagnostic studies (PK1)	8.705 [1]	8.333 [1]	8.622 [2]
Pharmacotherapy (PK2)	8.409 [3]	8.222 [3]	8.270 [4]
Differential diagnosis (PK3)	8.545 [2]	8.278 [2]	8.757 [1]
Observational reassessment (PK4)	8.045 [4]	8.028 [4]	8.622 [2]

**Table 5 healthcare-11-00471-t005:** The competency analysis of the PS aspect.

Staffs Criteria	PGYs	Residents	Visiting Staff
	(Staff A)	(Staff B)	(Staff C)
Horizontal analysis			
Emergency stabilization (PS1)	8.136 [3]	8.528 [2]	8.838 [1]
Clinical assessment (PS2)	8.750 [1]	8.139 [3]	8.622 [2]
Procedure management (PS3)	7.818 [3]	8.500 [1]	8.297 [2]
Ultrasound use (PS4)	7.500 [2]	8.028 [1]	7.270 [3]
Vertical analysis			
Emergency stabilization (PS1)	8.136 [2]	8.528 [1]	8.838 [1]
Clinical assessment (PS2)	8.750 [1]	8.139 [3]	8.622 [2]
Procedure management (PS3)	7.818 [3]	8.500 [2]	8.297 [3]
Ultrasound use (PS4)	7.500 [4]	8.028 [4]	7.270 [4]

**Table 6 healthcare-11-00471-t006:** The competency analysis of the PL aspect.

Staffs Criteria	PGYs	Residents	Visiting Staff
	(Staff A)	(Staff B)	(Staff C)
Horizontal analysis			
Task transition (PL1)	7.477 [3]	7.583 [2]	8.162 [1]
Professional ethics (PL2)	8.250 [2]	7.722 [3]	8.405 [1]
Interpersonal communication (PL3)	8.182 [2]	7.833 [3]	8.459 [1]
Performance improvement (PL4)	8.091 [2]	7.861 [3]	8.162 [1]
Vertical analysis			
Task transition (PL1)	7.477 [4]	7.583 [4]	8.162 [3]
Professional ethics (PL2)	8.250 [1]	7.722 [3]	8.405 [2]
Interpersonal communication (PL3)	8.182 [2]	7.833 [2]	8.459 [1]
Performance improvement (PL4)	8.091 [3]	7.861 [1]	8.162 [3]

**Table 7 healthcare-11-00471-t007:** The competency analysis of the CS aspect.

Staffs Criteria	PGYs	Resident	Visiting Staff
	(Staff A)	(Staff B)	(Staff C)
Horizontal analysis			
Teamwork (CS1)	8.045 [2]	7.722 [3]	8.324 [1]
Patient safety (CS2)	8.455 [1]	7.889 [3]	8.351 [2]
System management (CS3)	7.932 [1]	7.361 [3]	7.784 [2]
Technology applications (CS4)	7.386 [2]	7.167 [3]	7.595 [1]
Vertical analysis			
Teamwork (CS1)	8.045 [2]	7.722 [2]	8.324 [2]
Patient safety (CS2)	8.455 [1]	7.889 [1]	8.351 [1]
System management (CS3)	7.932 [3]	7.361 [3]	7.784 [3]
Technology applications (CS4)	7.386 [4]	7.167 [4]	7.595 [4]

**Table 8 healthcare-11-00471-t008:** Component matrix with a rotated axis.

			Components	
Aspects	Components	Criteria	1	Community
Professional knowledge (PK)	Diagnostic and therapeutic knowledge (PKP1)	Diagnostic studies (PK1)	0.915	0.838
		Pharmacotherapy (PK2)	0.911	0.830
		Differential diagnosis (PK3)	0.897	0.805
		Observational reassessment (PK4)	0.852	0.726
	Eigenvalue λ	3.200		
	% of Variance	79.993		
	Cumulative (%)	79.993		
	Cronbach’s α	0.916		
Professional skills (PS)	Stabilization and procedural skills (PSP1)	Emergency stabilization (PS1)	0.874	0.764
		Procedure management (PS3)	0.847	0.717
		Ultrasound use (PS4)	0.820	0.673
		Clinic assessment (PS2)	0.739	0.547
	Eigenvalue λ	2.700		
	% of Variance	67.506		
	Cumulative (%)	67.506		
	Cronbach’s α	0.838		
Professional literacy (PL)	Professionalism and interpersonal communication (PLP1)	Professional ethics (PL2)	0.909	0.826
		Interpersonal communication (PL3)	0.897	0.805
		Task transition (PL1)	0.888	0.788
		Performance improvement (PL4)	0.875	0.765
	Eigenvalue λ	3.184		
	% of Variance	79.599		
	Cumulative (%)	79.599		
	Cronbach’s α	0.914		
Care services (CS)	System management & teamwork (CSP1)	System management (CS3)	0.939	0.881
		Teamwork (CS1)	0.927	0.859
		Patient safety (CS2)	0.846	0.716
		Technology applications (CS4)	0.840	0.706
	Eigenvalue λ	3.162		
	% of Variance	79.038		
	Cumulative (%)	79.038		
	Cronbach’s α	0.911		

**Table 9 healthcare-11-00471-t009:** The calculated weights of four components.

Aspects	Components	Component Weights
Professional knowledge (PK)	Diagnostic studies & Pharmacotherapy (PKP1)	0.257
Professional skills (PS)	Emergency stabilization & Management (PSP1)	0.259
Professional literacy (PL)	Professional ethics & communication (PLP1)	0.238
Care services (CS)	Care management & Teamwork (CSP1)	0.246
	Total	1.000

**Table 10 healthcare-11-00471-t010:** The CDI (competency development indicators) under *v* = 0, 0, 5, and 1.0.

		PGYs	Residents	Visiting Staff
		(Staff A)	(Staff B)	(Staff C)
*v* = 0.0	*R_vk_*	0.205	0.247	0.199
	CDI	0.795	0.753	0.801
	Rank	2	3	1
*v* = 0.5	*R_vk_*	0.197	0.225	0.185
	CDI	0.803	0.775	0.815
	Rank	2	3	1
*v* = 1.0	*R_vk_*	0.189	0.204	0.171
	CDI	0.811	0.796	0.829
	Rank	2	3	1

**Table 11 healthcare-11-00471-t011:** The rank of aspects/components and strategy adoption paths of competency development for the EPs.

	Advantage and Disadvantageous Aspects/Components	Strategy Adoption Paths (SAPs)
PGYs (Staff A)	Professional knowledge (PK) > Professional skills (PS) > Professional literacy (PL) > Care services (CS)	1. PL(3)→PS(2) {N} 2. PL(3)→PK(1)→PS(2) {Y} 3. PL(3)→CS(4)→PS(2) {Y} 4. PL(3)→CS(4)→PK(1)→PS(2) {Y}
Residents (Staff B)	Professional skills (PS) > Professional knowledge (PK) > Professional literacy (PL) > Care services (CS)	1. PL(3)→PS(1) {N} 2. PL(3)→PK(2)→PS(1) {N} 3. PL(3)→CS(4)→PS(1) {Y} 4. PL(3)→CS(4)→PK(2)→PS(1) {Y}
Visiting staff (Staff C)	Professional knowledge (PK) > Professional literacy (PL) > Professional skills (PS) > Care services (CS)	1. PL(2)→PS(3) {Y} 2. PL(2)→PK(1)→PS(3) {Y} 3. PL(2)→CS(4)→PS(3) {Y} 4. PL(2)→CS(4)→PK(1)→PS(3) {Y}
Common strategy adoption paths (Common SAPs)		3. PL→CS→PS 4. PL→CS→PK→PS

## Data Availability

The dataset presented in this research is available with a legitimate request from the corresponding author.

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
