# Peer review of "Exploring the Critical Driving Forces and Strategy Adoption Paths of Professional Competency Development for Various Emergency Physicians Based on the Hybrid MCDM Approach"

_healthcare, 2023, doi:10.3390/healthcare11040471_

Round 1

Reviewer 1 Report

Manuscript title: Exploring the Critical Driving Forces and Strategy Adoption Paths of Professional Competency Development for Various Emergency Physicians Based on the Hybrid MCDM Approach.

In this study, authors evaluate the professional competency of physicians' needs and establish competency development strategies. This study looked for differences in awareness and performance from the three types of emergency physicians, including attending physicians, emergency residents, and doctors in post-graduate years. For this need, the authors used multi-criteria decision-making methods: DEMATEL-ANP-VIKOR. The manuscript is of very high quality. It is well organized, and the contents fit with the journal’s topics. The case study is well-conceived. The applied methodology is well described and applied. The essence of this paper is in the case studies. I congratulate the authors on their very high-quality paper.

However, it also presents several flaws that need to be addressed before considering it for publication.

- Add the words „ANP” to the list of keywords.

- The flowchart of the methodology is missing.

- After the equations, describe the variables used in them.

- Is it possible to compare the obtained results with some earlier research?

Reviewer 2 Report

Dear Authors, Thanks for sharing your work. Overall, it is an interesting study, but I don´t believe it is presented as a paper. I felt like reading extracts of a thesis. I might be wrong, yet I believe this could be improved. Your introduction section is very long, and it does not point out what you are going to do in you study. I saw more like an introduction in your materials and methods section, which again is very long. I believe you should reorganise the information. Shorted the actual introduction and include there the pieces that are in the method section which should go in the introduction. Then in the method section you can just leave what you actually did and how you did it. I did not see the questionnaires you used, please make sure to include at least an extract of those. Also, there are many references need and others that are in the list but not used in the text. Please make sure that all ideas are supported and clean your list from those not present. The results section is a bit hard to read, with many acronyms that make you get lost. Please re consider simplifying this section and leave only your findings. The discussion section was week in comparison with all the rest of the text. Here you have the chance to compare your results with other finding in the literature. Please take a look there too. 

I think you have great material that should be presented in a better way. Hope my words help. 

Kind regards.

Reviewer 3 Report

The manuscript reports about a very complex study that "analyzes EPs’ professional competency needs and competency development status. It identifies four aspects (professional knowledge, professional skills, professional literacy, and care services) of competency development based on expert interviews and a literature review [...] analyzes professional competency and determine the driving forces of critical competency development [...] determines the components' weights [...] determines the EPs' critical influence factors for healthcare"

This is the main issue of the study: it is 3-4 different studies put together. The manuscript is hence too long, lacking methodological details when needed and filled with mathematical details which seem not appropriate for a medical journal. In the following I justify with some examples this judgement. I understand that this could sound a harsh judgement, but I believe that all the work the authors did deserves a better outcome and my goal is to suggest possible alternative options.

Lacking details
As far as I could understand, the basic empirical data which are the foundation of the study is the interview of a very limited set of experts (par. 2.1). There are no information on this very relevant part of the study, allowing replicability (questions of the interview? processing of data, according to one of the possible qualitative methods? how was literature integrated in the overall analysis?). This is a pity, because this part, properly developed, could be an article in itself: Table 1 could be an excellent final result! 

In Par. 2.2 the DEMATEL method is introduced, with a very long set of examples ( 25 lines!!) and only at the end of the paragraph the reader discovers the steps of the method ("(1) find the original average matrix, (2) calculate the direct influence matrix, (3) calculate the indirect influence matrix, (4) evaluate the total influence matrix, and (5) obtain the NRM (network relation map)". In the following line, the reader is informed that "Respondents respond to the influence effect...". Who are the respondents? The seven experts? Do the authors mean that the whole very long computations that follows are enterely based of the few figures suggested by 7 experts?
Maybe I completely misuderstood the study, but even in this case, this means that the manuscript needs to be clearer: if you consider me as the "mean reader", the clarity of the manuscript can be easily improved.
An experimental validation of the framework depicted in Table 1 could be a second very good article.

Too many details
Very often the description of the concepts is unclear and constructs are buried under a long list of references. The sections 2.1.1 and following do not clarify "what" are the four constructs, only providing a description of the process of healthcare at the ED. Each section should start with a positive definition of the construct. By the way, PC was defined as Patient Care at page 3, according to ACGME. It seems to become PK in the following.

Formulas: it is enough to quote the methods and their reference, there is no use in showing the way in which the various indexes are calculated (we don't do it for ordinary medical statistics, like univariate or multivariate analysis), unless you introduced something new in the formulas. But Healthcare is a medical journal ...

The sections in which you validate the model in Table 1, define PCA and suggest some educational consequences could easily be transformed into two more articles.

I hope this can be useful. Despite my overall positive judgement on the work you did, the manuscprit in its present form is not suitable for pubblication.

Reviewer 4 Report

Abstract

It is too long.

Mind you write in line 20, CMBE instead of CBME

We do need information about where the study was done (which country) and we need to know the study technique (online questionnaire survey), I think that the approach is less important.

There is no conclusion in the abstract.

Introduction

“In 2012, the ACGME launched the Next Accreditation System with 88 milestones in the core competency framework”. Add reference here.

Lines 120-121 “By analyzing competencies for physicians with different levels of seniority, we will integrate these professional competencies and design a  complete evaluation and training system.” This is quite an ambitious aim for the short paper.

Lines 126 – 137 are something that I have already read (should not this be a part of the methods section?)

Lines 137 – 143 should be rather part of the discussion section.

Lines 144 – 151 are unnecessary.

The article needs a clear purpose. A clear research question. I also don't understand why so many approaches were used in a short article.

I also did not find information about Taiwan, medical education there, and the approach to developing competences. This is important if then doctors from Taiwan are examined.

Materials and Methods

How many respondents from post-graduate years (PGYs), emergency residents, and attending physicians did you survey? And how many of those groups work in at Chang Gung Memorial 156 Hospital? What was the response rate?

First, you write you have done the survey, and then a very long (167 – 365) description of core competencies for EPs follows. I would recommend moving it to the annex (167 – 365), and living in this section just necessary information about research and questionnaire, as it is usually done in Healthcare papers.

Is the database from the study publicly available?

Round 2

Reviewer 3 Report

The manuscript is a revised version of a previous text. It has been widely changed and the main concerns raised in my previous review have been addressed.

There are still some minor typos (for example page 4 lines 151-52 "There are severe experts on the committee... I guess it is seven) and minor check to the English language ("by" is often used instead of "with", when a tool is used)

Figure 1 is very important, because it clarifies the rather complex flow of different elaborations, in which the reader could miss the sense of the research.

Nevertheless, beyond the technicalities of the method of analysis of the questionnaires, the whole research remains rather obscure in its sense, because the authors do not take an explicit theoretical educational position. I try to argue about this judgement.

As far as I can understand, the complex set of statistical elaborations is meant to transform the result of a questionnaire (it would be interesting for the reader to be able to read it) into a set of pathways of priorities, summarized with the predominance of the PL aspect. To my eyes, Figure 2 is a very good final result of the research. What I miss is an educational theory providing a key of interpretation to these findings and allowing to transform them into an educational approach. When the authors state that "By strengthening the aspect of PL, the CS, PK, and PS aspect can be improved sequentially" (page 21, lines 667-8) they make an assumption that is not based on empirical data of effectiveness, but on empirical data derived from a self-administered questionnaire. This statement is neither based nor declared coherent with a theoretical educational background.
I guess that the approach is very behaviouristic, because in reading the text I had the feeling that the authors believe that by moving a pawn on the chessboard, you can determine the outcome of the game in a mechanical way. I recommend more prudence in the final considerations.

The study deals with adult education (PG, residents, staff), where a lot of different factors are in action (intrinsic and estrinsic motivation, previous experience, educational environment, hidden cirriculum). The management of a hospital can decide to start with the improvement of PL, but well beyond the "content", what matters is the educational method and the learning environment. In adult education one size cannot fit all and the community of learners must decide their own pathway. This is not a "truth", but my pedagogical creed, based on social constructivism.

Finally, since the job of a reviewer is not to agree or disagree with what the authors wrote, but to check for internal inconsistencies and methodological errors, I have no reasons to reject this manuscript, but I'd like to ask the authors to add at least a disclaimer in the Limitations, remebering that:
- this study is based on a self-administered questionnaire and not on a pre-post study, showing the effectiveness of the interventions
- the proposed strategy is rooted in a behaviouristic (or maybe cognitivistic?) theoretical framework, that does not take into account the role of social educational interactions of each specific community of learners, which could have another preferred/more effective pathway 

Reviewer 4 Report

Thank you for correction.

Author Response

Thank you very much for the detailed and profound reading of the manuscript, as well as for your time and willingness to improve the quality and clarity of the paper.